# Factors that affect the utilisation of maternal healthcare in the Mchinji District of Malawi

**Catherine Louise Stewart[ORCID], Jennifer Anne Hall***

Reproductive Health Research Department, UCL Elizabeth Garrett Anderson Institute for Women's Health, London, United Kingdom

* jennifer.hall@ucl.ac.uk

## Abstract

### Background

It is widely accepted that maternal healthcare is vital for improving maternal and neonatal mortality rates. Furthermore, the continuum of care–the integrated delivery of antenatal, delivery and postnatal care–has been shown to be particularly important. Sub-Saharan Africa has the highest neonatal and maternal mortality rates in the world; significant improvements in the provision and utilisation of the continuum are urgently needed, therefore the barriers preventing access need to be better understood. This study aimed to identify key factors associated with the utilisation of maternal healthcare, in the Mchinji District of Malawi.

### Methods

4,244 pregnant women from the Mchinji District of Malawi were interviewed between March and December 2013. The overall utilisation of maternal healthcare was calculated by combining the use of antenatal, delivery and postnatal care into one variable—continuum of care. Univariate and multivariate logistic regressions were performed to determine the factors associated with utilisation of maternal healthcare.

### Results

Utilisation of maternal healthcare in the Mchinji District was inadequate; only 24% of women received the recommended package. Being further from a healthcare facility (OR = 0.2, 95%CI = 0.04–0.96), having at least one live child (OR = 0.87, 95%CI = 0.84–0.99), previous experience of miscarriage (OR = 0.64, 95%CI = 0.50–0.82) or abuse (OR = 0.81, 95%CI = 0.69–0.95) reduced utilisation, whereas being in the richest 20% (OR = 1.33 95%CI = 1.08–1.65), having a planned pregnancy (OR = 1.3, 95%CI = 1.11–1.51) or more control over decisions (OR = 1.09, 95%CI = 0.80–1.49) increased utilisation.

### Conclusion

Seven groups of women were identified as having an increased risk of low utilisation of maternal healthcare; women living >5km from a healthcare facility, within the poorest socio-economic group, experiencing an unplanned pregnancy, with at least one live child,

**Data Availability Statement:** The data underlying the results presented in the study are available from the UCL repository, DOI. 10.5522/04/19188617.

**Funding:** The study from which these data were drawn was funded by a three-year personal Research Training Fellowship from the Wellcome Trust to Dr J Hall, award number 097268/Z/11/Z. https://wellcome.org/ The funders had no role in study design, data collection and analysis, decision to publish, or preparation of the manuscript.

**Competing interests:** The authors have declared that no competing interests exist.

experience of a previous miscarriage, no control over their healthcare decisions or experience of abuse. Policy makers should pay extra attention to these high-risk groups when designing and delivering strategies to improve maternal healthcare utilisation.

## Introduction

It is widely accepted that the utilisation of effective maternal healthcare is key to improving both maternal and neonatal mortality rates [1]. Studies from around the world have shown that women who receive antenatal care, delivery care and postnatal care from skilled healthcare personnel have better pregnancy outcomes resulting in a reduction in both neonatal and maternal mortality rates [2–9]. The continuum of care for reproductive, maternal, newborn and child health, as defined by Kerber et al. in 2007, involves the integrated delivery of healthcare for both mothers and children from pre-pregnancy all the way through to childhood [10]. However, the continuum of care is often narrowed to concentrate on antenatal, delivery and postnatal care provided by skilled healthcare workers [11]. Although it is accepted that each element of care will improve outcomes, an integrated approach has been shown to be the most effective [10, 12, 13], and utilisation of the continuum of care is recognised as an important means of reducing maternal and neonatal mortality rates [10, 14, 15].

Sub-Saharan Africa has the highest neonatal and maternal mortality rates in the world, with 27 neonatal deaths per 1,000 live births [16] and 533 maternal deaths per 100,000 live births, accounting for approximately 68% of all maternal deaths globally [17]. Malawi, a sub-Saharan African country with a total population of just over 19.6 million [18], is one of the poorest countries in the world with a GDP per capita of only 642.7 $USD [19]; neonatal mortality and maternal mortality rates are approximately 19 deaths per 1,000 live births [20] and 451 per 100,000 live births respectively [21]. Although these mortality rates are not the highest of all sub-Sharan African countries, they are still regarded as unacceptably high and effective strategies are needed to significantly reduce these figures [21, 22].

While previous studies have highlighted the importance of maternal healthcare [2–9] and identified some of the factors that influence the individual utilisation of antenatal care, delivery care and postnatal care [23–31], very few studies have investigated the continuum of care. Given the importance of the integrated use of maternal healthcare, it is vital that the barriers preventing women accessing the continuum are understood [10, 12–15]. This information is key for policy makers to be able to design and implement effective strategies to increase utilisation of maternal healthcare across the continuum. This study investigated the key factors associated with the utilisation of maternal healthcare across the continuum in the Mchinji District of Malawi. By adding to the body of research investigating the utilisation of maternal healthcare, it sheds some much-needed light on the factors that are associated specifically with the utilisation of the continuum of care.

## Materials and methods

### Ethics statement

The study from which these data were drawn was approved by the UCL Research Ethics Committee and the College of Medicine Research Ethics Committee at the University of Malawi, reference numbers 3974/001 and P.03/12/1273 respectively, and the research was conducted in line with relevant guidelines and regulations. All participants gave written informed consent to participate, by thumbprint if necessary, after they had read the information sheet and/or

had the study explained to them. Both ethics committees approved this consent procedure. Local approval to conduct the research in Mchinji District was given by the District Health Officer and the District Executive Committee.

## Study setting and design

Malawi is a southeast sub-Saharan African country, bordered by Zambia, Mozambique and Tanzania. Malawi is separated into three regions: Northern, Central and Southern. Within the Central region is the Mchinji District; a rural district covering an area of 3,131 km$^2$ with a population of just over 600,000 [32]. Previous research has divided the Mchinji District into 49 areas of approximately equal populations [33]. From these areas 25 were randomly selected for participation in research on pregnancy intention and outcomes and all pregnant women, aged 15 and over, in the study area were identified and invited to participate [34, 35]. Overall, 4,244 pregnant women within the Mchinji District were interviewed between March and December 2013. They were interviewed in their homes by one of the 25 local trained data collectors [34, 35]. Each woman was interviewed twice, once whilst pregnant and once after giving birth.

This study is a secondary data analysis, investigating the integrated utilisation of antenatal care, delivery care and postnatal care by combining these three components of maternal health services into one outcome variable: continuum of care. Within this study, utilisation of antenatal care was measured by whether a woman attended four or more antenatal care appointments during her pregnancy or not; four visits was used as the cut off as, at the time of data collection, this was the number of appointments recommended by the World Health Organisation (WHO) [36]. Utilisation of delivery care was split between whether a woman delivered within a medical facility or not. Utilisation of postnatal care was measured by whether a woman received postnatal care within seven days of giving birth or not; seven days was used as the cut off for postnatal care as, at the time of data collection, this was the general recommendation [37]. Women were classified as either receiving the limited package (none or one of the three maternal healthcare services used), the moderate package (two of the three maternal healthcare services used) or the recommended package of maternal healthcare (all three of the maternal healthcare services used).

## Independent variables

The literature was scoped to identify previous research that had investigated factors that affect the utilisation of maternal healthcare in low- and middle-income countries (LMICs) (Table 1). The findings of these studies were then used to identify key factors that had previously been associated with the utilisation of maternal healthcare and 18 variables were chosen for inclusion in this study.

## Statistical analysis

To assess the determinants of utilisation of maternal healthcare univariate ordered logistic regressions of the chosen variables were performed to investigate the relationship between the variable and the utilisation of maternal healthcare. Multivariate ordered logistic regression was then used with the continuum of care variable as the outcome measure. A conceptual hierarchical model for the multivariate analysis was constructed made up of five levels (Fig 1). Each variable was assigned to a level based on the literature regarding the relationship between the variable and the utilisation of maternal healthcare as well as chronological considerations. Only variables that were found to be significant (p<0.1) in the univariate analysis were considered for inclusion in the multivariate model. All the variables within a level were introduced to the model simultaneously. Any variables found to be non-significant (p>0.1) when added into

**Table 1. Published literature and findings of multivariate analyses investigating the utilisation of maternal healthcare in LMICs.**

| Study | Sample Size | Factors in analysis | Main Results |
|---|---|---|---|
| **Chakraborty et al., 2003 [23] Determinants of the use of maternal health services in rural Bangladesh.** | 993 women. Data from survey of maternal morbidity in Bangladesh, conducted by BIRPERHT (1992–3). | Women's age, mother's education, number of previous pregnancies, family size, husband's occupation, women's employment status, SES, type of housing, distance to health facility. | Mother's education significantly effects utilisation of maternal health care, and is independent of other women's characteristics, SES and access to healthcare. |
| **Rahman et al., 2012 [38] Intimate partner violence and use of reproductive health services among married women: evidence from a national Bangladeshi sample.** | 2,001 women. Data from the 2007 Bangladesh Demographic Health Survey. | Mother's age, mother's education, husband's education, decision-making autonomy, employment, residence, religion, wealth quintile, media exposure, parity, pregnancy intention, intimate partner violence. | Women with experience of physical or sexual intimate partner violence were less likely to receive ANC and delivery care from a skilled provider. |
| **Birmeta et al., 2013 [25] Determinants of maternal health care utilization in Holeta town, central Ethiopia.** | 422 women. Cross section survey (2012). | Women's age, women's education, women's occupation, marital status, parity, household income, media exposure, pregnancy intention, knowledge of danger signs of pregnancy, husband's approval of care, decision making. | 87% of women had at least one ANC visit. Age at last birth, literacy status, family income, media exposure, attitude towards pregnancy, knowledge of danger signs and husband approval were all shown to be significant factors affecting ANC utilisation. While, parity, literacy status, family income, media exposure, autonomy over healthcare decisions, perception of distance and ANC utilisation were shown to significantly affect delivery care utilisation. |
| **Rai et al., 2013 [39] Factors Associated With the Utilization of Maternal Health Care Services Among Adolescent Women in Malawi.** | 2,160 women. Data from the fourth wave of the 2010 Malawi Demographic and Health Survey. | Women's age, women's education, husband's education, women's work status, women's personal barrier index, media exposure, wealth quintile, religion, ethnic group, head of the household, birth order and interval, pregnancy intention, residence, region. | Age, SES and pregnancy intention were shown to significantly affect ANC utilisation, while birth order and interval, religion and ethnicity affected PNC utilisation. |
| **Joshi et al., 2014 [26] Factors associated with the use and quality of antenatal care in Nepal.** | 4,079 women. Data from the 2011 Nepal Demographic and Health Survey. | Women's age, women's education, women's work status, wealth quintile of household, religion, smoking status, decision making power, media exposure, parity, pregnancy intention, previous obstetric experience, family planning, ecological zone, residence, husband's education, husband's occupation. | 85% of women had at least one ANC visit, with 50% having four or more. ANC utilisation increased with increasing age, education, SES and parity. |
| **Ononokpono & Odimegwu, 2014 [40] Determinants of Maternal Health Care Utilization in Nigeria: a multilevel approach.** | 17,542 women Data from the 2008 Nigeria Demographic and Health Survey. | Maternal age at last birth, education, religion, ethnic origin, occupation, women's autonomy, parity, SES, place of residence and region of residence. | Region of residence was associated with facility delivery; women living in Northern Nigeria less likely to have facility delivery than women residing in Southern Nigeria. |
| **Sialubanje et al., 2014 [27] Understanding the psychosocial and environmental factors and barriers affecting utilization of maternal healthcare services in Kalomo, Zambia.** | 176 women. 35 in-depth interviews, 12 focus groups. | Mother's age, distance to healthcare facility, number of live children, perceived quality of care, cost associated with care, control over decisions. | Low perception of healthcare services as well as physical and economic barriers were the main reasons for low utilisation of maternal healthcare. |
| **Singh et al., 2014 [28] Utilization of maternal healthcare among adolescent mothers in urban India: evidence from DLHS-3.** | 3,315 women. Data from the third round of the District Level Household Survey (2007–2008). | Woman's education, husband's education, religion, caste, exposure to mass media, wealth index, women's employment, parity, region of residence. | 22.9% of women received full ANC, 70.5% had a delivery assisted by skilled personnel and 65.1% had at least one PNC check-up. Education, religion, caste, SES, parity, exposure to healthcare messages and region of residence all affected ANC utilisation. While, education, ANC utilisation, SES, religion and region of residence affected safe delivery. PNC utilisation was associated with SES, education, ANC utilisation, safe delivery utilisation and region of residence. |

(*Continued*)

**Table 1.** (Continued)

| Study | Sample Size | Factors in analysis | Main Results |
|---|---|---|---|
| **Tsawe et al., 2015 [29]** **Factors influencing the use of maternal healthcare services in Swaziland.** | 4,987 women. Data from the 2006–07 Swaziland Demographic and Health Survey. | Women's education, woman's age, women's employment and income, SES, residence, parity, distance to health facilities, and exposure to the media. | 97.3% of women used ANC, 74% used delivery care, and only 20.5% used PNC. Age, parity, media exposure, maternal education, SES and residence were shown to affect utilisation of maternal healthcare. |
| **Haider et al., 2017 [30]** **Effects of women's autonomy on maternal healthcare utilization in Bangladesh.** | 8,753 women. Data from the 2011 Bangladesh Demographic Health Survey. | Women's autonomy, women's age, women's education, husband's age, husbands' education, birth order, religion, media exposure, residence, wealth quintile. | Women with no education, of Islamic faith, from poorest wealth quintile, living in rural areas and with low autonomy utilised maternal healthcare the least. Increase in autonomy increased utilisation of maternal healthcare. |
| **Kazanga et al., 2019 [41]** **Predictors of Utilisation of Skilled Maternal Healthcare in Lilongwe District, Malawi.** | 1,126 women. Data from the 2010 Malawi Demographic and Health Survey (MDHS). | Women's age, marital status, residence, education, work status, SES, ethnicity and religion. | Residence, education and SES all significantly affected utilisation of maternal healthcare, with women with less education, low SES or from urban areas less likely to utilise the continuum of maternal healthcare. |
| **Nuamah et al., 2019 [31]** **Access and utilization of maternal healthcare in a rural district in the forest belt of Ghana.** | 720 women. Cross section survey (2015). | Women's age, women's education, religion, marital status, employment status, number of live children, household wealth, health insurance, proximity to health facility, breastfeeding, use of family planning, preference of healthcare, knowledge about pregnancy-related conditions and danger signs. | 68.5 of women has >3 ANC visits, 83.6% had skilled delivery care and 33.6% used PNC services. Socio-economic characteristics and access to healthcare facility all influenced the utilisation of maternal healthcare. Increasing age increased ANC and PNC utilisation, while education and religion also significantly affected ANC utilisation. |
| **Oh et al., 2020 [42]** **Factors associated with the continuum of care for maternal, newborn and child health in The Gambia: a cross-sectional study using Demographic and Health Survey 2013.** | 1,308 women Data from the 2013 Gambia Demographic and Health Survey (GDHS). | Women's age, pregnancy intention, woman's autonomy in decision-making, health insurance, geographical location, distance to health facilities, women's education, husband's education, wealth quintile, media exposure, sex of the newborn, size of the newborn at birth, birth order, timing of the first antenatal check, knowledge of signs of pregnancy complications. | Autonomy over healthcare decisions, partner's education, listening to the radio, birth order, knowledge of signs of pregnancy complications, residency and distance to health facility were all shown to significantly affect the utilisation of the continuum of maternal healthcare. |

(SES–socio-economic status, ANC–antenatal care, PNC–postnatal care).

the model were removed using manual backwards stepwise elimination, starting with the variable with the highest p-value. Once a variable had been retained within the model it was not subsequently removed, even if it later became non-significant following the addition of new variables in the lower levels.

## Inclusivity in global research

Additional information regarding the ethical, cultural, and scientific considerations specific to inclusivity in global research is included in the S1 File.

## Results

### Background characteristics

Over 99% of women who were eligible for inclusion chose to participate, which suggests that the data collected are representative of the overall population of pregnant women within the Mchinji District of Malawi [35].

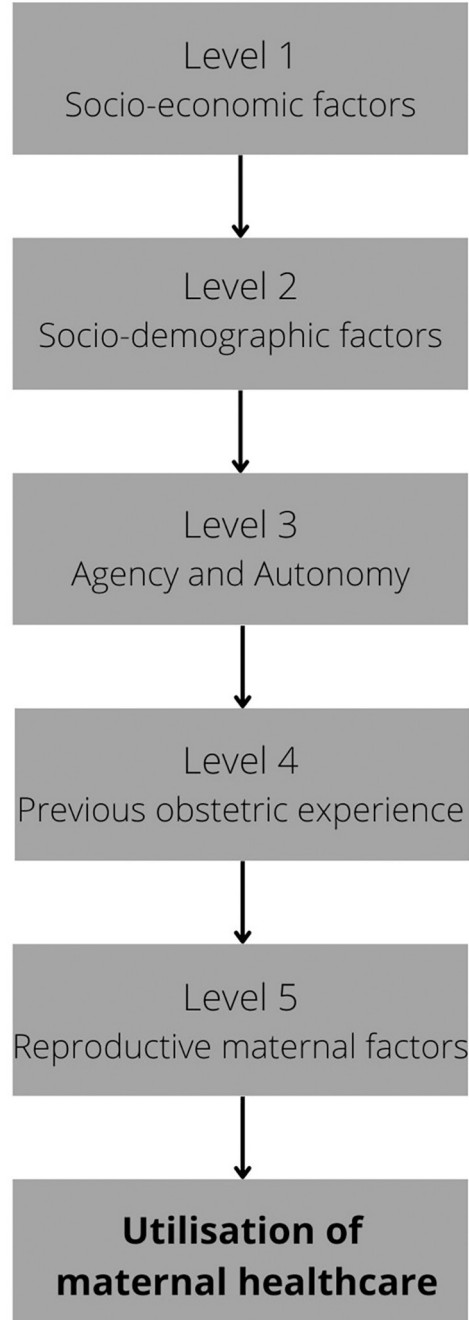

**Fig 1. Conceptual hierarchy for the determinants of the utilisation of maternal healthcare.**

The socio-demographic characteristics of the 3,986 women who completed both antenatal and postnatal interviews are shown in Table 2 (and have been published previously in detail) [35]. Briefly, almost all women were married (91.9%), ranged in age from 15–49 years with a median age of 24 years and the majority were only educated to primary level (75.7%). The fathers were aged between 15 and 71 years with a median age of 28 years and most men were also only educated to primary level (64%). 53% of the participants lived between 5 and 10km away from a healthcare facility. The socio-economic status (SES) of each woman was

**Table 2. Socio-demographic characteristics of women (and their partners).**

| | Mother | | Father | |
|---|---|---|---|---|
| | Frequency | Percent | Frequency | Percent |
| **Age** | | | n = 3,822 | |
| 15–19 | 954 | 23.9% | 132 | 3.5% |
| 20–29 | 2031 | 51% | 1985 | 55.4% |
| 30+ | 1001 | 25.1% | 1705 | 44.6% |
| Range (median) | 15–49 (24) | | 15–71 (28) | |
| **Education** | | | n = 3,922 | |
| None | 396 | 9.9% | 317 | 8.0% |
| Primary | 3018 | 75.7% | 2508 | 64% |
| Secondary | 562 | 14.1% | 1079 | 27.5% |
| Tertiary | 10 | 0.3% | 18 | 0.5% |
| **Marital Status** | | | | |
| Married | 3,665 | 91.9% | | |
| Unmarried | 321 | 8.1% | | |
| **Distance to HCF** | | | | |
| 0-5km | 1449 | 36.4% | | |
| 5-10km | 2115 | 53.1% | | |
| 10-15km | 410 | 10.3% | | |
| >15km | 7 | 0.2% | | |
| **Range (average)** | 0.1–15.8 (5.9) | | | |

calculated based on assets (such as ownership of a bicycle or radio), household characteristics (such as the materials on the floor or roof), household density, and access to sanitation facilities and water. An ordered categorical variable for SES (wealth quintile) was then calculated using the forementioned factors, and all the women were split into one of five groups, ranging from richest 20% to poorest 20% [32, 35].

The obstetric history of the 3,986 women is shown in Table 3 (also previously published in detail) [35]. Briefly, the median number of pregnancies (including the current pregnancy) was three, while the number of previous births, both live and still, ranged from zero to 12. 71% of participants had had a previous pregnancy and of those women, 16.5% had reported at least one previous miscarriage, 7.4% reported at least one still birth and 27% reported at least one child death. The median number of live children was one, but the number ranged from zero to nine. Just over a third of women (35%) had previously given birth within the last 24 months, though the median birth interval was three years.

4,244 women completed the first, antenatal interview during their pregnancy and 3,986 women went on to complete the second, postpartum interview, one month after giving birth. Overall, there was a loss to follow up of 258 women (6.08%). The main reason for why women were lost to follow-up was due to migration, with only 27 women not consenting to the postnatal interview [35]. Loss to follow-up is a common problem in cohort studies and can cause a significant problem [35]. However, when the 258 women in this study who were lost to follow up were compared to the other 3,986 women there were no statistically significant differences between the two groups with the only exception being that the women lost to follow up were shown to be, on average, slightly younger (mean age was 24.0 years versus 25.1 years, p<0.01) [35]. Due to the relatively small loss to follow up and the lack of statistically significant differences, it is reasonable to assume that the postnatal sample is still representative of the overall population of pregnant women within the Mchinji District of Malawi [35].

**Table 3. Obstetric history of women.**

| | Frequency | Percent |
|---|---|---|
| **Number of Pregnancies** | | |
| First | 1,081 | 27.1% |
| 2-3<sup>rd</sup> | 1,315 | 33.0% |
| >4 | 1,590 | 39.9% |
| Range (median) | 1–15 (3) | |
| **Number of previous births (live and still)** | | |
| None | 1,145 | 28.7% |
| 1–2 | 1,337 | 33.6% |
| >3 | 1,504 | 37.7% |
| Range (median) | 0–12 (2) | |
| **Time Since Last Birth** | | |
| First birth | 1,154 | 28.95 |
| <24 months | 974 | 34.4% |
| 1–2 years | 827 | 29.2% |
| 3–4 years | 516 | 18.2% |
| 4–5 years | 257 | 9.1% |
| 5 years | 258 | 9.1% |
| Range (median) | 7–264 (36) | |
| **Number of live children** | | |
| 0 | 1250 | 31.3% |
| 1 | 806 | 20.2% |
| 2 | 605 | 15.2% |
| 3 | 540 | 13.6% |
| 4 | 394 | 9.9% |
| >5 | 391 | 9.8% |
| Range (median) | 0–9 (1) | |
| **Previous Miscarriage** | | |
| None | 3,507 | 87.9% |
| 1 | 369 | 9.3% |
| >2 | 110 | 2.8% |
| Range (median) | 0–6 (0) | |
| **Previous Stillbirth** | | |
| None | 3,772 | 94.6% |
| 1 | 187 | 4.7% |
| >2 | 27 | 0.7% |
| Range (median) | 0–3 (0) | |
| **Previous Child Death** | | |
| None | 3,197 | 80.2% |
| 1 | 563 | 14.1% |
| >2 | 226 | 5.7% |
| Range (median) | 0–6 (0) | |

## Univariate analysis

Following the univariate analysis, the factors shown to have a significant effect on the utilisation of maternal healthcare were; mother's age, father's age, mother's education, father's education, SES, pregnancy intention, birth interval, number of live children, previous miscarriage, previous stillbirth, previous child death, distance to healthcare facility, control over decisions,

abuse ever, abuse while pregnant and sexual abuse. Marital status and abuse in the last year were not significant (Table 4).

Distance to healthcare facility showed the most striking association with utilisation of maternal healthcare, showing that as distance from healthcare facility increased, the utilisation of maternal healthcare decreased (Fig 2).

## Multivariate analysis

All variables with a p-value of <0.10 in the univariate analysis were included in the multivariate hierarchical analysis. Marital status and abuse in last year were automatically excluded as they were not significant in the univariate analysis. All the other variables were included and were organised into one of five levels: socioeconomic factors, socio-demographic factors, agency and autonomy, previous obstetric experience, and maternal reproductive factors (Table 5).

Table 6 shows the final version of Model 5 of the multiple linear regressions. Model 5 shows that SES is associated with increased utilisation of maternal healthcare and shows that the effect is not mediated through the other variables lower down the hierarchy. Distance to healthcare facility is associated with a decrease in the utilisation of maternal healthcare and is not mediated through agency and autonomy variables, previous obstetric experiences, or maternal reproductive factors. Experience of abuse ever is associated with a decrease in maternal healthcare utilisation, while increasing level of control, in general, is associated with an increase in utilisation, even after controlling for SES and socio-demographic factors, suggesting that these effects are not mediated through other factors in the hierarchy. The effect of experience of sexual abuse, once included in the model, becomes of borderline statistical significance. This suggests that the effect seen is mediated through other factors, such as control over decisions. Previous miscarriage is associated with a decrease in maternal healthcare utilisation and is not mediated by maternal reproductive factors. Number of live children is also associated with a decrease in utilisation while increasing pregnancy intention is associated with an increase, after controlling for SES, socio-demographic factors, agency and autonomy variables and previous obstetric experiences.

Abuse while pregnant was dropped from Model 3, previous stillbirth and previous child death were dropped from Model 4 and mother's age and birth interval were dropped from Model 5. Mother's education, father's age and father's education all became insignificant following the addition of other variables in the hierarchy which means that their effect is likely mediated through agency and autonomy variables, previous obstetric experiences or maternal reproductive factors.

Overall, the multivariate analysis suggests that the key determinants of utilisation of maternal healthcare were SES, distance to healthcare facility, previous miscarriage, experience of abuse, control over decisions, number of live children and pregnancy intention.

## Discussion

Only 23% of women received the recommended package of maternal healthcare; four or more antenatal care visits, facility delivery, and postnatal care within 7 days of giving birth. Inspection of the factors shown to be associated with the utilisation of maternal healthcare by multivariate analysis led to the identification of seven groups of women who are at high risk of reduced utilisation. These include any women who fall within one of the following seven categories: living over 5km from a healthcare facility, falling within the poorest socio-economic group, experiencing an unplanned pregnancy, experience of a previous miscarriage, having at least one live child, having no control over healthcare decisions, or having experience of abuse.

**Table 4. Univariate ordered logistic regression results.**

| Variable | Odds Ratio | 95% Confidence Interval | | P-Values |
|---|---|---|---|---|
| **Mother's Age** | | 15–19 as baseline | | |
| 20–29 | 0.69 | 0.59 | 0.80 | <0.001 |
| 30+ | 0.55 | 0.47 | 0.66 | |
| **Mother's Education** | | None as baseline | | |
| Primary | 1.39 | 1.14 | 1.70 | <0.001 |
| Secondary | 2.18 | 1.70 | 2.79 | |
| Tertiary | 4.71 | 1.41 | 15.8 | |
| **Father's Age** | | 15–19 as baseline | | |
| 20–29 | 1.06 | 0.75 | 1.49 | <0.001 |
| 30+ | 0.77 | 0.55 | 1.09 | |
| **Father's Education** | | None as baseline | | |
| Primary | 1.22 | 0.98 | 1.52 | <0.001 |
| Secondary | 1.69 | 1.33 | 2.15 | |
| Tertiary | 4.23 | 1.69 | 10.6 | |
| **SES** | | Poorest as baseline | | |
| Poorer | 1.40 | 1.15 | 1.69 | <0.001 |
| Middle | 1.37 | 1.13 | 1.65 | |
| Richer | 1.26 | 1.04 | 1.52 | |
| Richest | 1.60 | 1.32 | 1.94 | |
| **Pregnancy Intention** | | Unplanned as baseline | | |
| Ambivalent | 1.12 | 0.95 | 1.32 | <0.001 |
| Planned | 1.44 | 1.25 | 1.65 | |
| **Birth Interval** | | Within 24 months as baseline | | |
| 2–3 years | 1.02 | 0.86 | 1.22 | <0.001 |
| 3–4 years | 1.38 | 1.12 | 1.69 | |
| 4–5 years | 1.48 | 1.13 | 1.93 | |
| 5+ years | 1.41 | 1.08 | 1.85 | |
| **Distance to HCF** | | 0-5km as baseline | | |
| 5-10km | 0.55 | 0.49 | 0.63 | <0.001 |
| 10-15km | 0.49 | 0.40 | 0.61 | |
| >15km | 0.21 | 0.05 | 0.92 | |
| **Control over Decisions** | | None as baseline | | |
| Few | 1.32 | 1.13 | 1.55 | <0.001 |
| Some | 1.36 | 1.11 | 1.67 | |
| Most | 1.66 | 1.35 | 2.02 | |
| All | 1.01 | 0.76 | 1.34 | |
| **Marital Status** | 0.97 | 0.78 | 1.21 | 0.802 |
| **No. of Live Children** | 0.87 | 0.84 | 0.99 | <0.001 |
| **Previous Stillbirth** | | No history of stillbirth as baseline | | |
| History of stillbirth | 0.95 | 0.73 | 1.25 | <0.001 |
| No previous pregnancy | 1.63 | 1.42 | 1.87 | |
| **Previous Miscarriage** | | No history of miscarriage as baseline | | |
| History of miscarriage | 0.81 | 0.67 | 0.98 | <0.001 |
| No previous pregnancy | 1.58 | 1.37 | 1.81 | |
| **Previous Child Death** | | No history of child death as baseline | | |
| History of child death | 0.89 | 0.76 | 1.04 | <0.001 |
| No previous pregnancy | 1.58 | 1.37 | 1.82 | |

(*Continued*)

**Table 4.** (Continued)

| Variable | Odds Ratio | 95% Confidence Interval | | P-Values |
|---|---|---|---|---|
| **Abuse Ever** | 0.72 | 0.62 | 0.83 | <0.001 |
| **Abuse in last year** | 0.87 | 0.69 | 1.10 | 0.245 |
| **Abuse while pregnant** | 0.63 | 0.46 | 0.87 | 0.005 |
| **Sexual Abuse** | 0.58 | 0.38 | 0.90 | 0.015 |

The majority of the findings from the univariate analyses are consistent with the published literature. However, mother's age, father's age, previous miscarriage, previous stillbirth and previous child death all showed an opposite relationship to the predicted hypothesis.

As each of these groups may overlap, a woman could potentially fall into all seven groups. Extra consideration should be paid to each of these groups of women when policy makers are designing and delivering strategies to improve utilisation of maternal healthcare.

Following multivariate analysis, distance to healthcare facility was shown to have one of the most striking associations with utilisation of maternal healthcare; women living 5-10km away from a healthcare facticity were 40% less likely to utilise the recommended package of maternal healthcare compared to women who live less than 5km away (OR 0.60, 95% CI 0.52–0.68). Furthermore, women who live over 15km away were 80% less likely to utilise the recommended package compared to those living less than 5km away (OR 0.20, 95% CI 0.04–0.88). This finding is consistent with published literature; distance has consistently been recognised as a key determinant of utilisation of maternal healthcare [27, 42–44]. Previous studies have suggested that this reduction is due, in part, to a lack of transport, the time taken to reach the facility and the costs involved [27, 29, 44]. In Malawi over 80% of the population live in rural

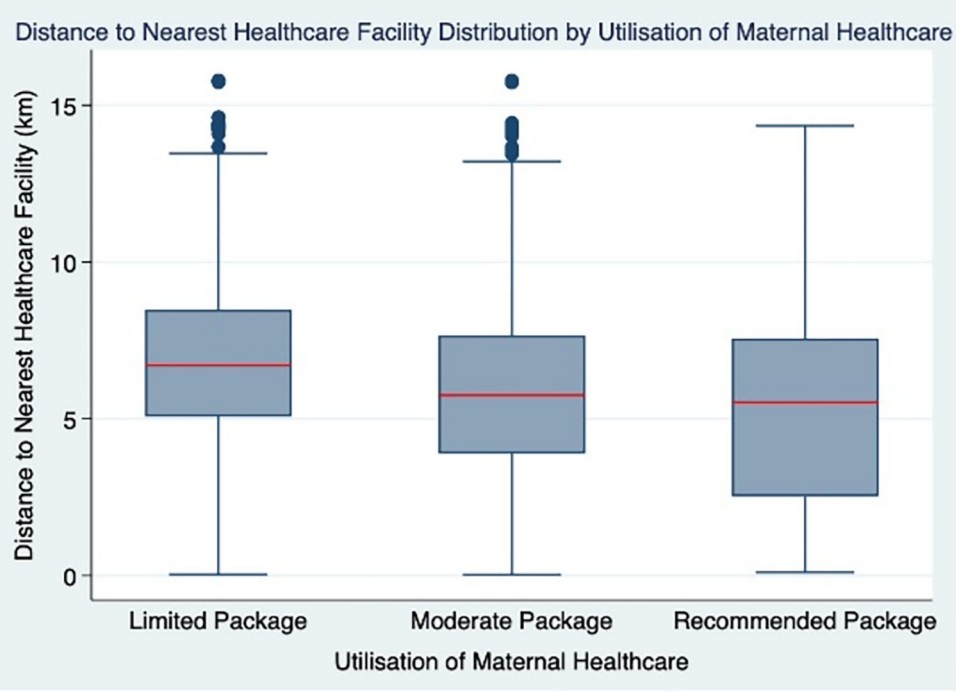

**Fig 2. Box plot showing median and inter-quartile range of distance to nearest healthcare facility by utilisation of maternal healthcare.**

**Table 5. Variables considered at each level of the hierarchal model.**

| Level | Variables |
|---|---|
| Level 1 | Asset Index |
| • Socio-Economic Factors | |
| Level 2 | Women's Education |
| • Socio-Demographic Factors | Father's Education |
| | Father's Age |
| | Distance to HCF |
| Level 3 | Abuse Ever |
| • Agency and Autonomy | Abuse while pregnant |
| | Sexual Abuse |
| | Control over decisions |
| Level 4 | Previous miscarriage |
| • Previous Obstetric Experiences | Previous stillbirth |
| | Previous child death |
| Level 5 | Mother's Age |
| • Maternal Reproductive Factors | Birth interval |
| | Pregnancy intention |
| | Number of live children |

areas, where most of the roads are dirt roads making travel difficult [45]. Additionally, the main method of travel is public transport, which is often unreliable and costly [46]. The multivariate analysis also showed that SES is a key factor with women in the lowest socio-economic group having a significantly lower utilisation of maternal healthcare compared to women in the higher socio-economic groups. This is an unsurprising finding, supported as it is by a vast amount of previous research [26, 29, 31, 41, 47, 48]. SES has also been linked to multiple other variables within the hierarchical model. For example, there is a clear link between SES and distance to healthcare facility and the subsequent utilisation of maternal healthcare; not only does SES dictate where people live, but it also affects the individual's ability to access healthcare due to the costs associated, such as travel [46].

The number of live children was shown to influence the utilisation of maternal healthcare with the addition of each live child decreasing the utilisation; this finding is again consistent with the published literature [23, 28, 29, 49–52]. It is thought that women who have had previous children rely on experience and knowledge from these previous pregnancies and births, and so are less likely to use maternal healthcare services [28, 53, 54]. Having live children may also cause restraints on resources, such as money or time, which could also result in reduced utilisation [55, 56]. The degree of pregnancy intention was also shown to be an important determinant of the utilisation of maternal healthcare; women with unplanned pregnancies were shown to have a reduced utilisation of the recommended package of maternal healthcare compared to women with planned pregnancies. This finding is in line with the results from previous studies [11, 57–60]. Unplanned pregnancies have been linked to an inadequate access to contraceptives and family planning services [61], therefore it is likely that women who are experiencing an unplanned pregnancy would also have reduced access to other aspects of maternal healthcare [61]. Also, a woman with an unplanned pregnancy may not realise she is pregnant or may not want others to know about the pregnancy, and so may be less likely to utilise maternal healthcare, especially antenatal care [59, 61].

Women who have control over decisions that affect their health had an increased utilisation of the recommended package of maternal healthcare. The results show that, in general, as

**Table 6. Model 5: SES, socio-demographic factors, agency and autonomy, previous obstetric experience, maternal reproductive factors, and utilisation of maternal healthcare.**

| | Odds Ratio | 95% CI | | p-value |
|---|---|---|---|---|
| **Socio-economic status** | Poorest as baseline | | | |
| Poorer | 1.42 | 1.16 | 1.74 | 0.008 |
| Middle | 1.32 | 1.08 | 1.61 | |
| Richer | 1.21 | 1.00 | 1.47 | |
| Richest | 1.33 | 1.08 | 1.65 | |
| **Mother's Education** | None as baseline | | | |
| Primary | 1.09 | 0.87 | 1.36 | 0.238 |
| Secondary | 1.29 | 0.97 | 1.72 | |
| Tertiary | 1.92 | 0.55 | 6.73 | |
| **Father's Education** | None as baseline | | | |
| Primary | 1.02 | 0.80 | 1.29 | 0.335 |
| Secondary | 1.16 | 0.88 | 1.51 | |
| Tertiary | 1.62 | 0.62 | 4.22 | |
| **Father's Age** | 15–19 as baseline | | | |
| 20–29 | 1.07 | 0.72 | 1.55 | 0.930 |
| 30+ | 1.07 | 0.72 | 1.61 | |
| **Distance to HCF** | 0-5km as baseline | | | |
| 5-10km | 0.60 | 0.52 | 0.70 | <0.001 |
| 10-15km | 0.55 | 0.44 | 0.68 | |
| >15km | 0.20 | 0.04 | 0.96 | |
| **Abuse Ever** | 0.81 | 0.69 | 0.95 | 0.008 |
| **Sexual Abuse** | 0.64 | 0.41 | 1.01 | 0.054 |
| **Control** | None as baseline | | | |
| Few | 1.32 | 1.12 | 1.55 | <0.001 |
| Some | 1.23 | 0.99 | 1.52 | |
| Most | 1.74 | 1.40 | 2.16 | |
| All | 1.09 | 0.80 | 1.49 | |
| **Previous Miscarriage** | No previous history of miscarriage as baseline | | | |
| History of miscarriage | 0.84 | 0.69 | 1.03 | 0.002 |
| No previous pregnancy | 1.32 | 1.09 | 1.60 | |
| **Number of live children** | 0.94 | 0.89 | 0.99 | 0.038 |
| **Pregnancy Intention** | Unplanned as baseline | | | |
| Ambivalent | 1.11 | 0.94 | 1.32 | 0.003 |
| Planned | 1.30 | 1.11 | 1.51 | |

control increases so does the utilisation of the recommended package. This finding is in line with previous research [25, 26, 38, 42, 62]. The results also show that women who have experienced abuse at some point in their life are less likely to utilise maternal healthcare compared to women who have never experienced abuse. This result fits with previous studies that have also shown that abuse (specifically intimate partner violence) is linked to reduced utilisation of maternal healthcare [38, 63–65]. It is thought that this may be due to the link between intimate partner violence and autonomy; women that experience abuse tend to have limited control over decisions, reduced freedom to travel and often higher economic dependency, all of which are also linked to a reduced utilisation of maternal healthcare [66–68]. There is also the possibility that women who experience abuse may refrain from utilising maternal healthcare services due to fears of exposing their abuse, perhaps due to shame or fear of consequences [69].

Finally, both the univariate and multivariate analyses showed that women who had experienced a previous miscarriage had a reduced chance of utilising the recommended package of maternal healthcare compared to women without experience of a miscarriage. This result was the opposite to the expected hypothesised result given that the majority of the research shows that the experience of a previous miscarriage increases the utilisation of maternal healthcare, often due to increased anxiety of subsequent fetal loss [70–74]. However, most of this research is in developed countries, such as the United States and Sweden. By contrast there is very little research into the effect that previous miscarriage has on the utilisation of maternal healthcare in low-income countries, such as Malawi. One potential explanation for the relationship seen is that women who do not utilise maternal healthcare have a higher chance of experiencing adverse pregnancy outcomes such as miscarriage, and women who have not utilised maternal healthcare in previous pregnancies are less likely to use it in subsequent pregnancies [75]. Therefore, women who have experienced a miscarriage are likely to have had a low utilisation of maternal healthcare and those women will continue to have a low utilisation, likely due to other factors, such a distance to healthcare facility.

The multivariable analysis highlights seven high-risk groups of women to whom policy makers should pay particular attention to when designing and delivering strategies aimed at improving the utilisation of maternal healthcare. For example, our findings show that there needs to be an increased focus on improving access to healthcare facilities. Our analyses identified distance to healthcare facility as a key determinant of utilisation and previous studies have cited lack of transport as one of the main barriers [44, 76]. Community-organised transport schemes have been shown to be effective in increasing maternal healthcare utilisation by providing transport to women within the community, enabling them to access healthcare facilities more easily. One example of an effective transport initiative is the provision of subsidised travel through the distribution of transport vouchers, enabling women to access free transport to healthcare facilities for antenatal, delivery and postnatal care [77]. Improvements to existing public transport as well as road development would also enable more reliable and more widely available transport, allowing more women to access healthcare facilities [78]. However, our analyses also showed that women with reduced control over healthcare decisions had reduced utilisation, suggesting that the people who have control over the decisions, usually a partner or mother-in-law, underestimate the importance of maternal healthcare. Therefore, it is vital that any scheme or intervention aimed at improving maternal healthcare not only targets women of reproductive age, but also those individuals within the community that have the control over healthcare decisions [79].

## Strengths and limitations of this study

One of the main strengths of this study is that the data used was collected during antenatal and post-partum follow ups, providing information about the current pregnancy and birth. As this study used data collected during the pregnancy in question, recall bias is far less of a concern. A further strength of the study is that due to the high response rate and low loss to follow up it can be inferred that the data is representative of the overall population of pregnant women in the Mchinji District [35].

A limitation of this study is that the utilisation of maternal healthcare for a woman whose pregnancy ended in a miscarriage during this study was still measured as utilisation of all three aspects of maternal healthcare. So, even though these women would not have needed delivery care they would have been classed as not utilising delivery care, rather than not needing it. This means that these women would automatically be classed in either the moderate or limited package. Despite this, the distribution of women within the three groups of healthcare

utilisation (limited, moderate and recommended) remained similar, regardless of whether women who miscarried were included or not. Another limitation is that the quality of the care received was not investigated, therefore we do not know if the care received was adequate. This would be an important area to study in the future as having high utilisation of maternal healthcare is only effective if the care being received is of a good quality.

## Conclusions

The findings from this study show that there was inadequate utilisation of maternal healthcare care along the continuum of care by women in the Mchinji District of Malawi. The key determinants associated with the utilisation of the continuum of maternal healthcare were distance to healthcare facility, SES, number of live children, pregnancy intention, previous miscarriage, control over decisions and experience of abuse. These findings should be taken into consideration by policy makers to design and deliver strategies that target women at a high-risk of low utilisation.

## Supporting information

**S1 File. Inclusivity in global research questionnaire.**
(DOCX)

## Acknowledgments

We would like to thank the MaiMwana Project with whom Dr Hall worked to establish this study, and LMUP team fieldworkers who collected the data used in this analysis, as well as all the women who consented to take part in the study.

## Author Contributions

**Conceptualization:** Catherine Louise Stewart, Jennifer Anne Hall.

**Data curation:** Catherine Louise Stewart, Jennifer Anne Hall.

**Formal analysis:** Catherine Louise Stewart, Jennifer Anne Hall.

**Funding acquisition:** Jennifer Anne Hall.

**Investigation:** Catherine Louise Stewart, Jennifer Anne Hall.

**Methodology:** Catherine Louise Stewart, Jennifer Anne Hall.

**Project administration:** Catherine Louise Stewart.

**Supervision:** Jennifer Anne Hall.

**Validation:** Catherine Louise Stewart, Jennifer Anne Hall.

**Visualization:** Catherine Louise Stewart.

**Writing – original draft:** Catherine Louise Stewart.

**Writing – review & editing:** Catherine Louise Stewart, Jennifer Anne Hall.

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
