## [Decision Letter · Decision Letter 0]

14 Sep 2022

PONE-D-22-20553Factors that affect the utilisation of maternal healthcare in the Mchinji District of Malawi.PLOS ONE

Dear Dr. Stewart,

Thank you for submitting your manuscript to PLOS ONE. After careful consideration, we feel that it has merit but does not fully meet PLOS ONE’s publication criteria as it currently stands. Therefore, we invite you to submit a revised version of the manuscript that addresses the points raised during the review process.

We look forward to receiving your revised manuscript.

Kind regards,

Rajesh Raushan, PhD

Academic Editor

PLOS ONE

Journal Requirements:

Reviewers' comments:

Reviewer's Responses to Questions

**Comments to the Author**

1. Is the manuscript technically sound, and do the data support the conclusions?

Reviewer #1: Yes

Reviewer #2: Yes

2. Has the statistical analysis been performed appropriately and rigorously? 

Reviewer #1: Yes

Reviewer #2: Yes

3. Have the authors made all data underlying the findings in their manuscript fully available?

Reviewer #1: Yes

Reviewer #2: Yes

4. Is the manuscript presented in an intelligible fashion and written in standard English?

Reviewer #1: Yes

Reviewer #2: Yes

5. Review Comments to the Author

Reviewer #1: The research work sounds a piece of scientific work, particularly for Malawi and similar setting. The table 1 presents good and clear picture of why some variables were chosen for further investigation. Although the percentages were explained in the text, but it would be good to present in table with background characteristics and N against each background characteristics.

In discussion section, the findings state that “… living 5-10km away is associated with 0.6 times the odds of…”, the interpretation of the odds ratio needs to refine.

Reviewer #2: The study was supposed to examine the utilisation of maternal health and various factors affecting the utilisation of these health services.

The paper is written in a very crisp form explaining the background, objectives, methodology, and conclusion. The method chosen for this particular study is completely justifying in the context of the objectives given. Table 2 onwards provided in the research paper clearly depicts what they are meant for. The discussion part of the paper is also written in a well-mannered.

Only two suggestions to improve or revise the paper are as follows-

1. Table 1 can be more elaborated in terms of the sample size and results in one or two lines.

2. The authors have mentioned that the findings can be taken into consideration for policy making. If it is possible to mention one or two major findings which are strongly needed for the policy imperatives will enhance the value of this paper.

The paper can be considered for publication.

6. PLOS authors have the option to publish the peer review history of their article (what does this mean?). If published, this will include your full peer review and any attached files.

Reviewer #1: No

Reviewer #2: **Yes: **Geeta Sahu

---

## [Author Response · Author response to Decision Letter 0]

10 Oct 2022

Reviewer #1: 

1. The research work sounds a piece of scientific work, particularly for Malawi and similar setting. The table 1 presents good and clear picture of why some variables were chosen for further investigation. - Thank you.

2. Although the percentages were explained in the text, but it would be good to present in table with background characteristics and N against each background characteristics. - We have added a table listing the background characteristics, showing the n value for each characteristic. We have also done the same for the obstetric history information. 

3. In discussion section, the findings state that “… living 5-10km away is associated with 0.6 times the odds of…”, the interpretation of the odds ratio needs to refine. - We agree that odds ratio can be hard to interpret, therefore, we have changed the wording of the findings in the discussion - it now discusses the findings in percentages, so is hopefully easier to understand (we have retained the odds ratios in brackets).

Reviewer #2: 

- The study was supposed to examine the utilisation of maternal health and various factors affecting the utilisation of these health services. The paper is written in a very crisp form explaining the background, objectives, methodology, and conclusion. The method chosen for this particular study is completely justifying in the context of the objectives given. Table 2 onwards provided in the research paper clearly depicts what they are meant for. The discussion part of the paper is also written in a well-mannered.

Only two suggestions to improve or revise the paper are as follows. - Thank you. 

1. Table 1 can be more elaborated in terms of the sample size and results in one or two lines. - We have updated Table 1, so it now includes the sample size and a brief summary of the results for each of the studies. We also removed one of the studies, replacing it with another similar study for which a precise sample size was available. 

2. The authors have mentioned that the findings can be taken into consideration for policy making. If it is possible to mention one or two major findings which are strongly needed for the policy imperatives will enhance the value of this paper. - We have added a section to the end of the discussion highlighting some suggestions for policy based off our results. As a result of this addition, we have slightly altered the order of points within the discussion to ensure the discussion flows well. Thank you for this suggestion – we agree that this change strengthens our paper.

Academic editor’s Comments

1. Please ensure that your manuscript meets PLOS ONE's style requirements, including those for file naming. - We have gone through and checked that our manuscript meets PLOS ONE’s style requirements and have named the files as directed.

- Edited the subheadings to be sentence case.

- Added supporting information section to end of manuscript

- Added DOIs to references

2. Please include a complete copy of PLOS’ questionnaire on inclusivity in global research in your revised manuscript. - A completed copy of PLOS ONE’s questionnaire on inclusivity in global research is included as a supporting information file - referenced in the main manuscript under a subsection ‘Inclusivity in global research’ within the Methods section.

3. Your ethics statement should only appear in the Methods section of your manuscript - We have added an ethics statement to the start of the Methods section.

4. Please review your reference list to ensure that it is complete and correct. - We have reviewed our reference list and can confirm that it is now complete and correct. We have made some changes to our references list 

- Replaced Ovikuomagbe, 2017 with Ononokpono et al., 2014 as detailed information regarding sample size was available.

- Removed four duplicate reference (Rai et al., 2013, Chakraborty et al., 2003, Singh et al., 2014, Tsawe et al., 2015).

- Removed Mubangizi, 2016 as poster presentation not publication (didn’t replace as already had other references saying similar things).

- Reordered the reference list to reflect the change in the order of discussion. 

- Added DOIs to the references 

5. While revising your submission, please upload your figure files to the Preflight Analysis and Conversion Engine (PACE) digital diagnostic tool. - We uploaded our figures to PACE and can confirm that they now meet the PLOS ONE’s requirements.

---

## [Decision Letter · Decision Letter 1]

12 Dec 2022

Factors that affect the utilisation of maternal healthcare in the Mchinji District of Malawi.

PONE-D-22-20553R1

Dear Dr. Stewart,

We’re pleased to inform you that your manuscript has been judged scientifically suitable for publication and will be formally accepted for publication once it meets all outstanding technical requirements, other than the minor comments raised by one of the reviewers.

Kind regards,

Rajesh Raushan, PhD

Academic Editor

PLOS ONE

Additional Editor Comments (optional):

Authors are advised to work on the minor comments raised by one of the reviewers along with the required editing. Afterwards, it can be accepted for the possible publication. 

Reviewers' comments:

Reviewer's Responses to Questions

**Comments to the Author**

1. If the authors have adequately addressed your comments raised in a previous round of review and you feel that this manuscript is now acceptable for publication, you may indicate that here to bypass the “Comments to the Author” section, enter your conflict of interest statement in the “Confidential to Editor” section, and submit your "Accept" recommendation.

Reviewer #1: All comments have been addressed

Reviewer #2: All comments have been addressed

2. Is the manuscript technically sound, and do the data support the conclusions?

Reviewer #1: Yes

Reviewer #2: (No Response)

3. Has the statistical analysis been performed appropriately and rigorously? 

Reviewer #1: Yes

Reviewer #2: Yes

4. Have the authors made all data underlying the findings in their manuscript fully available?

Reviewer #1: Yes

Reviewer #2: Yes

5. Is the manuscript presented in an intelligible fashion and written in standard English?

Reviewer #1: Yes

Reviewer #2: Yes

6. Review Comments to the Author

Reviewer #1: The preferable categories to show the socio-economic status are wealth quintile. If the present work has wealth status in quintile, it can also be shown in terms of "Poorest" which will show bottom 20% of the population followed by Poorer, Middle, Richer and Richest. The author can consult DHS reports for detail.

Similarly, education: if Respondent have 0 years of education, it can be categorized as Illiterate.

Reviewer #2: all the comments provided by the reviewer has been taken into the consideration and incorporated well.

7. PLOS authors have the option to publish the peer review history of their article (what does this mean?). If published, this will include your full peer review and any attached files.

Reviewer #1: No

Reviewer #2: No

---

## [Editor Report · Acceptance letter]

22 Dec 2022

PONE-D-22-20553R1 

Factors that affect the utilisation of maternal healthcare in the Mchinji District of Malawi. 

Dear Dr. Stewart:

I'm pleased to inform you that your manuscript has been deemed suitable for publication in PLOS ONE. Congratulations! Your manuscript is now with our production department. 

Kind regards, 

on behalf of

Dr. Rajesh Raushan 

Academic Editor

PLOS ONE